# BLSP: Bootstrapping Language-Speech Pre-training via Behavior Alignment of Continuation Writing

## Abstract

The emergence of large language models (LLMs) has sparked significant interest in extending their remarkable language capabilities to speech. However, modality alignment between speech and text still remains an open problem. Current solutions can be categorized into two strategies. One is a cascaded approach where outputs (tokens or states) of a separately trained speech recognition system are used as inputs for LLMs, which limits their potential in modeling alignment between speech and text. The other is an end-to-end approach that relies on speech instruction data, which is very difficult to collect in large quantities. In this paper, we address these issues and propose the **BLSP** approach that **B**ootstraps **L**anguage-**S**peech **P**re-training via behavior alignment of continuation writing. We achieve this by learning a lightweight modality adapter between a frozen speech encoder and an LLM, ensuring that the LLM exhibits the same generation behavior regardless of the modality of input: a speech segment or its transcript. The training process can be divided into two steps. The first step prompts an LLM to generate texts with speech transcripts as prefixes, obtaining text continuations. In the second step, these continuations are used as supervised signals to train the modality adapter in an end-to-end manner. We demonstrate that this straightforward process can extend the capabilities of LLMs to speech, enabling speech recognition, speech translation, spoken language understanding, and speech conversation, even in zero-shot cross-lingual scenarios.[1]

## 1 Introduction

Large Language Models (LLMs), trained on massive amounts of textual data, have achieved significant success on various natural language processing tasks (Chowdhery et al., 2022; OpenAI, 2023; Gao et al., 2023). Recent research efforts have attempted to expand LLMs' capabilities to comprehend diverse modalities (Yin et al., 2023; Latif et al., 2023). Speech, as an important modality, offers a plethora of benefits that complement text-based communication. Speech not only serves as the primary mode of human interaction but also conveys rich emotions, tones, and intentions that cannot be fully captured in text. Thus, enabling LLMs to understand speech could greatly enhance their utility in real-world scenarios.

However, effectively integrating and aligning speech with LLMs remains a significant challenge. Current approaches can be classified into two categories. One adopts a cascade paradigm, where the LLM is equipped with an automatic speech recognition (ASR) model to convert speech into text (Huang et al., 2023; Shen et al., 2023), or the LLM is fed output states from a separately trained recognition system (Chen et al., 2023). In this setup, the transfer of knowledge from the LLM to the speech modality is hindered due to the separation between ASR and LLM training. Recent efforts explore training end-to-end speech-language LLMs for direct speech interaction (Zhang et al., 2023; Shu et al., 2023). Yet, this approach heavily relies on scarce speech instruction data, which is challenging to collect in large quantities, and struggles to generalize robustly across languages and speakers. In this work, we address the question of whether it is possible to align speech and text in

---

[1]Video demos are available at `https://anonymous4blsp.github.io/iclr24-3521.github.io/`.

a generalized manner using existing cross-modal datasets like ASR data, which is available in large volumes.

Our preliminary investigation has found that a model trained to predict the ground-truth transcript with speech input loses the ability to follow instructions. To achieve effective cross-modal alignment, we introduce the BLSP approach, which bootstraps language-speech pre-training via behavior alignment of continuation writing. The key idea is to ensure that LLMs exhibit consistent behavior when prompted by speech or the corresponding text. Specifically, we first prompt an LLM to generate text continuations from speech transcripts. Then we use these continuations as supervised signals to train a modality adapter inserted between the speech encoder and LLM, by requiring the LLM to generate the same text continuations when prompted with the corresponding speech. Our experiments reveal that the BLSP approach can efficiently achieve cross-modal alignment, enabling LLMs to understand speech while retaining their language capabilities.

The contributions of our work are as follows:

- We introduce a novel approach to bootstrap language-speech pre-training through behavior alignment of continuation writing, providing a new direction for cross-modal alignment in LLMs.
- We develop a straightforward process that requires training only a lightweight modality adapter, leveraging a pretrained speech encoder and LLM, and utilizing existing speech recognition data, thus eliminating the need to acquire speech instruction data.
- We conduct quantitative evaluations and provide video demonstrations to showcase that our BLSP approach effectively extends LLMs to speech inputs, enabling speech recognition, speech translation, spoken language understanding, and speech conversation, even in zero-shot cross-lingual scenarios.

## 2 BACKGROUND

Due to the scarcity of speech instruction data, a natural approach to align speech and text for leveraging LLMs is to connect a pre-trained speech encoder to an LLM through a modality adapter trained on large volumes of speech-transcript pairs collected for speech recognition. Similar methods have achieved considerable success in vision-language models. Notably, BLIP-2 (Li et al., 2023) and MiniGPT-4 (Zhu et al., 2023) have demonstrated that training a learnable interface using aligned image caption data can effectively bridge the modality gap between vision and language, enabling an LLM to comprehend images while retaining its capacity to follow text prompts.

However, this approach proves to be more intricate when it comes to achieving speech and text alignment. Our preliminary investigation has found that the modality adapter trained to predict the ground-truth transcript from speech input can cause the LLM to only perform speech recognition with speech input. It completely ignores the textual instructions provided before the speech segment, regardless of the diverse transcription instructions employed during training[2]. We speculate that training on homogeneous ASR training data may result in a strong bias in the learned speech representations that confines the LLM to the transcription task only.

To substantiate our hypothesis, we conducted an analysis of the representations learned from ASR task on speech and text pairs from the LibriSpeech dataset (Panayotov et al., 2015). We consider four distinct tasks: continuation writing (CW), sentiment analysis (SA), speech recognition (SR), and speech translation (ST). For each task, the same instruction is employed to prompt the LLM to process either speech or its corresponding transcript. The cross-modal prompt format is as follows: *###[Human]:<task instruction><speech/transcript>\n\n\n###[Assistant]:*. The learned representations of these inputs are obtained by extracting the hidden state of the final layer for the last token of the cross-modal prompt, right before response generation. We provide a visualization of the representations of 25 samples in Figure 1. Ideally, paired speech and transcript inputs should yield similar representations, and these representations should be clustered based on task instructions. However, this visualization clearly demonstrates the separation between speech and text representations in the feature space. The LLM consistently projects a speech input into almost identical representations regardless of instructions provided, resulting in overlapping markings in the figure. This indicates

---

[2]We leverage OpenAI GPT-4 to generate 100 instructions for ASR task.

a lack of ability to adhere to instructions when processing speech inputs. To further illustrate this pattern, we present the average cosine similarity between the learned representations of the same input across different task instructions in Table 1. Notably, the representations for speech input are remarkably similar regardless of the task instruction used. Table 2 calculates the average cosine similarity between representations of paired speech and text inputs, considering the same task instructions. The consistently low similarity scores suggest a lack of alignment between the representations of speech and text inputs. This inability to bridge the modality gap through ASR tasks prompts us to reevaluate what it means to align speech and text for LLMs.

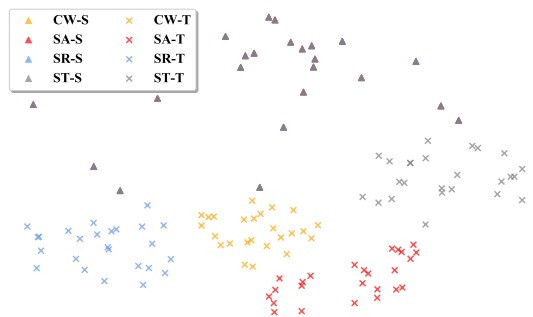

|       | CW-S  | SA-S  | SR-S  | ST-S  |
|-------|-------|-------|-------|-------|
| CW-S  | 1.000 | 0.997 | 0.997 | 0.991 |
| SA-S  | 0.997 | 1.000 | 0.997 | 0.992 |
| SR-S  | 0.997 | 0.997 | 1.000 | 0.993 |
| ST-S  | 0.991 | 0.992 | 0.993 | 1.000 |

Table 1: Average similarity between representations of the same speech inputs under different task instructions learned from ASR task.

| CW    | SA    | SR    | ST    |
|-------|-------|-------|-------|
| 0.270 | 0.106 | 0.328 | 0.176 |

Table 2: Average similarity between representations of paired speech/text inputs under the same task instructions learned from ASR task.

Figure 1: T-SNE visualization of feature representations learned from ASR task. Colors denote task instructions: orange for continuation writing (CW), red for sentiment analysis (SA), blue for speech recognition (SR), and gray for speech translation (ST). Shapes distinguish input modality: triangles for speech, crosses for text. Note that speech inputs result in overlapping features.

## 3 METHOD

Our proposed approach, named **B**ootstrapping **L**anguage-**S**peech **P**retraining (**BLSP**) via behavior alignment of continuation writing, is designed to align pre-trained speech encoders with large language models (LLMs), with the goal of extending the remarkable language capabilities to speech input. Our model comprises three components: a speech encoder, an instruction-following LLM, and a modality adapter between the speech encoder and LLM. We keep both the speech encoder and the LLM frozen during the training process and only train the parameters of the modality adapter. An overview of our model is presented in Figure 2. We will next describe how to construct data to train the modality adapter in an end-to-end manner.

### 3.1 BEHAVIOR ALIGNMENT OF CONTINUATION WRITING

Instead of treating the speech-transcript pairs as input-output mappings, we can utilize them from other perspectives. For example, we can consider the speech and its transcript in each pair as two independent inputs to the LLMs. Intuitively, if the representations of a speech segment are well aligned in the textual space for an LLM, the LLM should behave the same no matter whether it is given the speech segment or its transcript as input, i.e., it should generate the same text. This is behavior alignment. Among various behaviors, such as transcription, translation, and others, next token prediction may be the most universal, as evidenced by the success of LLMs. Following this universal next token prediction, we propose cross-modal behavior alignment of continuation writing, which utilizes LLM-generated continuations of speech transcripts to supervise the learning of the modality adapter in generating the same continuations when given the corresponding speech inputs.

This approach consists of two steps. In the first step, we prompt the LLM to continue writing after a speech transcript using the following instruction:

*###[Human]:Continue the following text in a coherent and engaging style with less than 40 words. <transcript>\n\n\n###[Assistant]:*

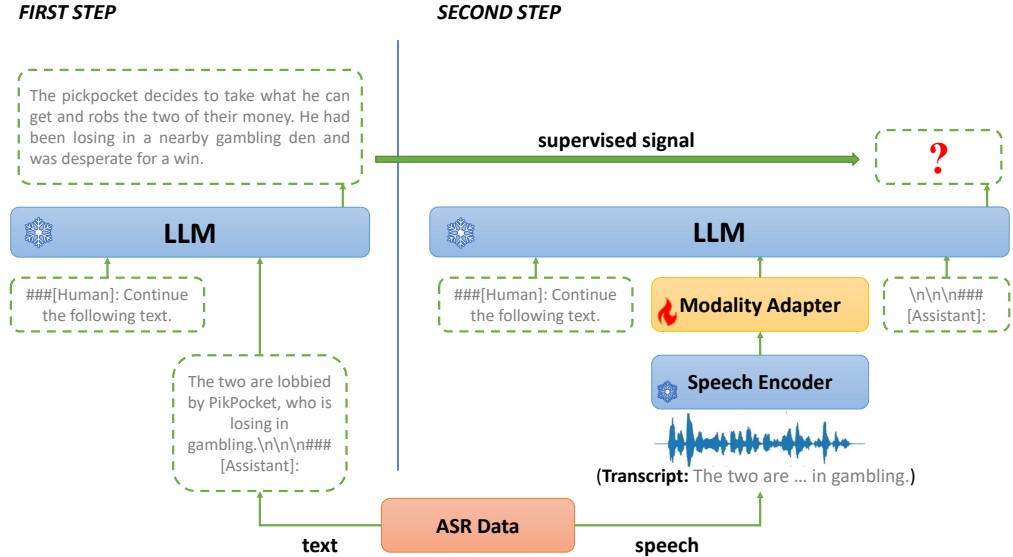

Figure 2: An overview of our BLSP approach for behavior alignment. Text continuations generated given speech transcripts as inputs by an LLM (in the first step) are used as supervisions to train the modality adapter given the corresponding speech as inputs (in the second step).

In the second step, we replace the word embeddings of the transcript with the speech features resulting from the modality adapter and regard the continuation produced in the first step as the ground truth. The model is optimized to generate the same continuations with the language modeling loss when provided with corresponding speech input, as prompted by the same instruction:

*###[Human]:Continue the following text in a coherent and engaging style with less than 40 words. <speech features>\n\n\n###[Assistant]:<text continuation>*

## 3.2 TRAINING DETAILS

We utilize Whisper-small (Radford et al., 2022) as the speech encoder and employ Llama-2-7B (Touvron et al., 2023) as the large language model (LLM). To induce instruction-following capability[3], we employ the publicly accessible dataset Alpaca-52K (Taori et al., 2023) to fine-tune the LLM. The Alpaca-52k dataset consists of 52K (text instruction, text input, response) triplets, which we convert into (text instruction, response) pairs by combining the instructions and inputs. During this stage, we fine-tune all parameters of the LLM for 3 epochs with a batch size of 128. This process takes about 2 hours using 8 A100 GPUs.

The modality adapter is composed of three 1-dimensional convolution layers followed by a bottleneck layer (Houlsby et al., 2019) with a hidden dimension of 512. The convolution layers are designed to reduce the length of the speech features by a factor of 8, with each layer having a stride size of 2, a kernel size of 5, and a padding of 2. To train the modality adapter, we utilize publicly available speech recognition datasets, including LibriSpeech (Panayotov et al., 2015), GigaSpeech (Chen et al., 2021), and Common Voice 2.0 (Ardila et al., 2020). We use the fine-tuned Llama-2 model to continue writing after speech transcripts, obtaining about 8.8M (speech, text continuation) pairs. During this stage, we fine-tune the modality adapter for 1 epoch with a batch size of 768. This process takes about 2.5 days using 8 A100 GPUs.

---

[3]We could have also used the official chat version of Llama-2, but we opted to perform instruction finetuning using publicly available data, as it offers flexibility for future research involving multi-modal instruction data.

## 4 EXPERIMENTS

We have found through experiments that the proposed BLSP approach is capable of empowering the LLM with speech understanding capabilities while maintaining fidelity to text instructions. We conduct evaluations on multiple speech-related downstream tasks, including speech recognition, speech translation, and spoken language understanding. It is important to note that the model is trained solely on the continuation writing task; therefore, all evaluations are conducted in a **zero-shot** setting, where we utilize text instructions to perform various speech-to-text generation tasks. Additionally, we demonstrate that our model supports cross-modal conversations and develops multilingual capabilities, even though the alignment training is carried out only in English.

### 4.1 QUANTITATIVE EVALUATIONS

**Speech Recognition**    We evaluate the speech recognition capabilities of our model quantitatively on both in-domain (LibriSpeech, Panayotov et al., 2015) and out-of-domain (TED-LIUM 3, Hernandez et al., 2018; Artie, Meyer et al., 2020) test sets. We use the prompt[4] *Please transcribe the following audio into English text.* to enable transcription and employ greedy search without sampling during generation. As evaluation metrics, we use Word Error Rate (WER) and BERTScore[5] (Zhang et al., 2019) for their ability to measure transcription accuracy and semantic similarity respectively.

| Method | In-Domain | | Out-of-Domain | |
|---|---|---|---|---|
| | LibriSpeech-dev | LibriSpeech-test | TED-LIUM 3 | Artie |
| whisper-small | 3.7 / 91.8 | 3.4 / 91.7 | 4.3 / 91.4 | 10.4 / 89.0 |
| BLSP | 20.6 / 87.5 | 21.7 / 87.2 | 33.1 / 83.5 | 31.3 / 84.8 |

Table 3: ASR results (WER / BERTScore) on different datasets.

Results in Table 3 demonstrate our model can transcribe from speech despite being trained on a text continuation task. However, there is a significant gap in transcription accuracy, as measured by WER, compared to specialized speech recognition systems represented by whisper-small, which shares the same speech encoder as our model. The gap in semantic similarity, as measured by BLEUScore, is relatively smaller. Some typical examples are highlighted in Table 4. While the BLSP model is able to perform the transcription task and can recognize most words in the speech, it is not as faithful. Moreover, the strong language generation ability of LLMs tends to produce more fluent transcripts, such as inserting prepositions missed in the speech or paraphrasing with less consideration of the original word order.

---

**Input:** <English Speech> Transcript: after early nightfall the yellow lamps would light up here and there the squalid quarter of the brothels
**BLSP:** After the early nightfall, the yellow lamps would light up here and there in the squalid quarter of the brothels.

---

**Input:** <English Speech> Transcript: a cold lucid indifference reigned in his soul
**BLSP:** A cold, lukewarm indifference reigned in his soul.

---

Table 4: Examples of speech recognition results.

**Speech Translation**    We then evaluate the speech translation capabilities of our model, which can be enabled by a translation instruction: *Please translate the following English audio into <target> text.* In this prompt, <target> represents the desired translation direction. We use SacreBLEU (Post, 2018) and COMET[6] (Rei et al., 2022) as evaluation metrics to measure lexical translation quality

---

[4]Note that we choose this particular prompt because it faithfully describes the transcription task. Other prompts could result in better transcription performance. For example, using the prompt *Please repeat the following words.* reduces WER scores to 17.0 on LibriSpeech-dev and 17.4 on LibriSpeech-test.

[5]The evaluation model is microsoft/deberta-xlarge-mnli

[6]The evaluation model is Unbabel/wmt22-comet-da

and semantic similarity, respectively. For comparison, we include cascaded results from ASR+LLM, where the speech recognition output of the whisper-small model is translated by the same fine-tuned Llama-2 model used in the BLSP model, and text translation results from Text+LLM, where the input to the LLM is the ground-truth speech transcripts. In both cases the LLM is prompted to perform text translation with the instruction: *Please translate the following English text into <target> text.*

| Method | en-ca | en-de | en-id | en-sl | en-sv |
|---|---|---|---|---|---|
| ASR+LLM | 19.2 / 69.1 | 17.3 / 70.5 | 15.8 / 75.2 | 10.0 / 65.1 | 22.2 / 74.5 |
| Text+LLM | 24.8 / 75.8 | 21.9 / 77.5 | 22.0 / 81.4 | 12.9 / 70.8 | 27.8 / 81.5 |
| BLSP | 14.1 / 64.0 | 14.3 / 66.3 | 12.0 / 71.4 | 6.9 / 59.6 | 16.8 / 68.9 |

Table 5: ST results (BLEU / COMET) on in-domain dataset CoVoST-2.

| Method | en-de | en-es | en-fr | en-it | en-nl | en-pt | en-ro | en-ru |
|---|---|---|---|---|---|---|---|---|
| ASR+LLM | 16.9 / 69.3 | 19.2 / 69.8 | 20.5 / 65.3 | 16.4 / 71.7 | 20.7 / 75.2 | 17.1 / 72.3 | 10.9 / 64.9 | 11.4 / 70.1 |
| Text+LLM | 20.2 / 73.4 | 21.7 / 74.1 | 24.2 / 69.7 | 20.5 / 76.3 | 24.4 / 80.0 | 20.3 / 76.7 | 13.3 / 68.6 | 13.3 / 74.5 |
| BLSP | 13.2 / 64.2 | 14.8 / 65.1 | 14.6 / 59.4 | 12.4 / 66.7 | 16.2 / 69.1 | 12.8 / 66.7 | 6.7 / 57.4 | 9.7 / 66.1 |

Table 6: ST results (BLEU / COMET) on out-of-domain dataset MUST-C.

Table 5 summarizes in-domain results on CoVoST-2 (Wang et al., 2020) for five translation directions: English (en) to Catalan (ca), German (de), Indonesian (id), Slovenian (sl), and Swedish (sv), for which our model achieves a MT BLEU score greater than 5. Table 6 presents out-of-domain results on MUST-C (Di Gangi et al., 2019) for all eight translation directions: English (en) to German (de), Spanish (es), French (fr), Italian (it), Dutch (nl), Portuguese (pt), Romanian (ro), and Russian (ru). As shown in these results, although there is a consistent gap in translation quality compared to the cascaded approach, our system is able to achieve reasonable translations across multiple translation directions, indicating that the proposed modality alignment approach extends the translation capability of LLMs to speech inputs.

**Spoken Language Understanding**   We also evaluate our model on spoken language understanding tasks, focusing on capturing the semantics expressed in the speech with less emphasis on the lexical level. The experiment is conducted on intent classification (IC) datasets SNIPS (Saade et al., 2019) and FSC (Lugosch et al., 2019), and sentiment analysis (SA) dataset SLUE-VoxCeleb (Shon et al., 2022). Similar to the ST evaluation, we include two alternative settings, ASR+LLM and Text+LLM, for comparison. The detailed instructions for each dataset can be found in Appendix A.

As illustrated in Table 7, our model demonstrates proficiency in spoken language understanding tasks when prompted by text instructions. While there is a performance drop on some datasets (SNIPS) when compared to the cascaded system ASR+LLM, our model outperformed the cascaded system on others (FSC and SLUE-VoxCeleb). This result indicates the potential of our approach in speech understanding.

| Method | IC | | | SA |
|---|---|---|---|---|
| | SNIPS-light-close | SNIPS-light-far | FSC | SLUE-VoxCeleb |
| ASR+LLM | 83.2 | 76.2 | 56.3 | 67.2 |
| Text+LLM | 86.3 | 86.3 | 72.4 | 71.6 |
| BLSP | 75.8 | 66.6 | 60.9 | 70.8 |

Table 7: SLU results (Accuracy for IC task, F1 score for SA task) on different datasets.

## 4.2 ANALYSIS

**Effectiveness as a Pre-Training Strategy**   We assess the effectiveness of the BLSP approach as a pre-training strategy for downstream tasks, focusing on speech translation as an example. Specifi-

cally, we utilize the same translation instruction as in zero-shot translation and fine-tune the parameters of the BLSP model to predict target translations from speech inputs, using training data for the eight translation directions from the MUST-C dataset. We employ LoRA (Hu et al., 2021) to adapt the key, query, value and output layers of the LLM's self-attention mechanism, with LoRA hyperparameters set to $R = 16$ and $\alpha = 16$. In addition, we update the parameters of the speech encoder and modality adapter. For comparison, we include alternative pre-trained models of the same architecture, but with a modality adapter that is either randomly initialized or pre-trained based on the ASR task utilizing the same datasets employed for BLSP training, as detailed in Section 3.2.

| Method | en-de | en-es | en-fr | en-it | en-nl | en-pt | en-ro | en-ru |
|---|---|---|---|---|---|---|---|---|
| w/o pretraining | 21.1 / 74.4 | 25.4 / 76.1 | 29.9 / 75.6 | 20.6 / 76.1 | 23.6 / 76.8 | 25.3 / 76.7 | 16.4 / 74.7 | 13.7 / 73.5 |
| ASR pretraining | 22.7 / 76.6 | **27.9** / 78.7 | **32.1** / 77.7 | 22.3 / 78.2 | 25.4 / 78.7 | 27.3 / 79.6 | 18.6 / 77.4 | 14.9 / 76.2 |
| BLSP | **23.3 / 77.7** | 27.4 / **79.5** | 31.9 / **78.5** | **23.2 / 79.0** | **26.4 / 80.0** | **28.5 / 80.4** | **19.2 / 78.6** | **15.6 / 77.3** |

Table 8: ST results (BLEU / COMET) of fine-tuned models on MUST-C.

As shown in Table 8, our BLSP model demonstrates a remarkable advantage in pre-training the modality adapter for the downstream speech translation task, obtaining substantial improvements compared to random initialization. Although pre-training the modality adapter with the ASR task is also helpful, the acquired bias could impede its generalization capability for downstream tasks. This is evident as our BLSP approach clearly outperforms ASR pre-training, achieving significantly higher COMET scores in all translation directions and higher BLEU scores in 6 out of 8 directions.

**Effectiveness in Speech-Text Alignment** We assess the efficacy of the BLSP approach in aligning speech and text inputs, following the methods described in Section 2. As depicted in Figure 3, the distribution of the learned representations of speech inputs is no longer distinct from that of text inputs, in contrast to the scenario trained with the ASR task, as shown in Figure 1. Instead, the representations of speech inputs now share the same distribution as text inputs, and the representations of paired speech and text inputs are in close proximity to each other, often overlapping. In Appendix B, we provide quantitative evidence demonstrating that our BLSP model is able to extract distinct representations of the same speech input for different instructions, and the representations extracted for paired speech and text inputs closely resemble each other when subjected to the same instructions. These findings suggest that our BLSP approach effectively aligns speech and text inputs within the same space, thereby extending the instruction-following capabilities of LLMs to speech inputs.

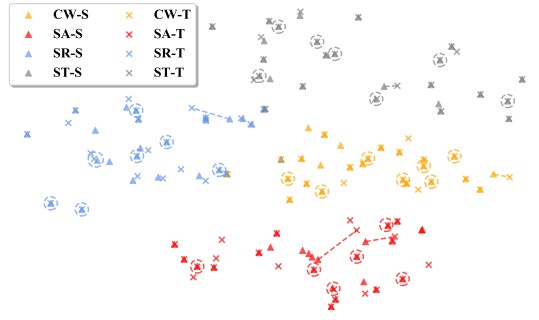

Figure 3: T-SNE visualization of feature representations learned from BLSP. Selected paired speech and text inputs are highlighted using dashed lines and circles.

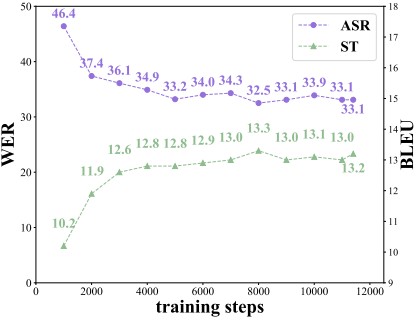

Figure 4: ASR and ST results at different training steps in one epoch.

**Impact of Data Size** We evaluate the impact of data size on model performance within the BLSP approach, utilizing measurements on out-of-domain datasets, specifically TED-LIUM 3 for zero-shot ASR performance and MUST-C en-de direction for zero-shot ST performance. In our experimental setup, we limit model training to a single epoch since the training loss converges well before the completion of one epoch. Consequently, we employ its performance at various training steps (approximately 0.8 million training samples for every 1,000 updates) as an estimate of its performance at different data scales. As shown in Figure 4, we observe rapid improvement in model

performance during the early stages of training, followed by convergence after approximately 5,000 updates (equivalent to around 4 million training samples).

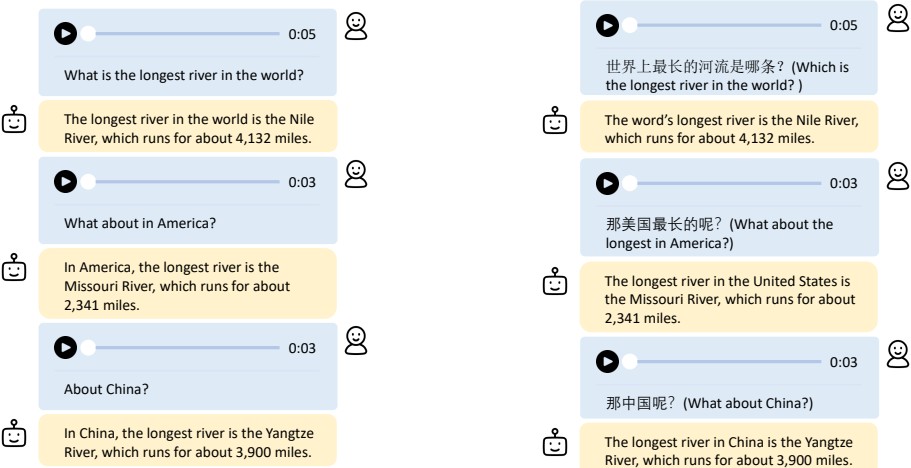

Figure 5: Speech conversation in English.    Figure 6: Speech conversation in Mandarin.

### 4.3 CROSS-MODAL CONVERSATION

We have observed that the BLSP approach can enable multi-turn conversation capabilities with LLMs using speech, thereby extending their remarkable conversational capabilities learned from text-only data to spoken languages. Figure 5 illustrates an example of engaging in a spoken conversation in English with the model. More examples are presented in Appendix C. Longer video demonstrations are available online[7].

### 4.4 EMERGENCE OF MULTILINGUAL CAPABILITIES

Despite being trained solely on English ASR data for behavior alignment in continuation writing, we have observed that the BLSP model demonstrates an understanding of non-English speech inputs. This can be attributed to the multilingual capabilities of both the speech encoder (Whisper, Radford et al. (2022)) and the LLM (Llama-2, Touvron et al. (2023)), as well as the specific design of the BLSP training process. Note that both the speech encoder and the LLM remain frozen during BLSP training, suggesting that despite training solely on English data, the modality adapter can learn to project the multilingual space in Whisper's output to the multilingual space for the LLM.

| Method | BSTC | MSLT | |
| --- | --- | --- | --- |
| | zh-en | de-en | fr-en |
| ASR+LLM | 11.1 / 54.3 | 25.3 / 79.4 | 24.1 / 71.5 |
| Text+LLM | 16.1 / 58.7 | 32.8 / 84.2 | 29.6 / 76.3 |
| BLSP | 5.0 / 49.8 | 13.1 / 70.9 | 13.4 / 64.8 |

Table 9: ST results (BLEU / COMET) in X-to-English directions.

To quantitatively measure the multilingual capabilities, we evaluate the speech translation performance of our model in the Chinese (zh) to English (en) direction on BSTC (Zhang et al., 2021) and in the German (de) and French (fr) to English (en) directions on MSLT (Federmann & Lewis, 2016). As shown in Table 9, the BLSP model demonstrates reasonable multilingual translation competency for source languages that were not observed during behavior alignment training. We note that there is a significant gap in translation quality, as measured by both BLEU and COMET, when compared to ASR+LLM and Text+LLM, suggesting room for improvement in future research.

---

[7]Video demos are available at `https://anonymous4blsp.github.io/iclr24-3521.github.io/`

As illustrated in Figure 6, our model is capable of engaging in multi-turn conversations with non-speech (Mandarin) speech input. It is worth mentioning that the model's responses are always in English. This is a direct result of the English-only training procedure in BLSP, where the continuations are consistently in English. This observation also suggests that there is benefit in incorporating multilingual training in behavior alignment for future research.

## 5 RELATED WORKS

**Multi-Modal Large Language Models** Current multi-modal large language models have been prominently focusing more on visual modality (OpenAI, 2023; Yin et al., 2023). These models utilize a pre-trained visual encoder to extract key visual features from images, which are then combined with text inputs to generate relevant outputs. PaLM-E (Driess et al., 2023) combines the huge 540B PaLM (Chowdhery et al., 2022) and the 22B Vision Transformer (ViT) (Dosovitskiy et al., 2020) to create the largest vision-language model currently reported. Since it would be costly to train a large multi-modal model in an end-to-end manner, many works introduce a learnable interface between the pre-trained visual encoder and LLM to connect different modalities while freezing the parameters of the pre-trained models. Flamingo (Alayrac et al., 2022), BLIP-2 (Li et al., 2023) and X-LLM (Chen et al., 2023) leverage a group of learnable query tokens to extract information in a query-based manner. LLaVa (Liu et al., 2023) connects the pre-trained CLIP (Radford et al., 2021) encoder and Vicuna (Chiang et al., 2023) with a simple projection layer. LLaMA-Adapter (Gao et al., 2023) and LaVIN (Luo et al., 2023) explore a parameter-efficient tuning manner, introducing a lightweight adapter module during training. Recent research has extended the above-mentioned approach to "audio" (Gong et al., 2023), which refers to natural sound, such as thunder and chirp. However, there is still a lack of exploration when it comes to human speech.

**Interact with LLMs through Speech** After the introduction of ChatGPT, several studies have focused on combining specialized speech models with LLMs, allowing for speech interaction with these language models. Initial endeavors in this field (e.g., HuggingGPT (Shen et al., 2023), AudioGPT (Huang et al., 2023)) employed a cascading model structure, linking LLMs with additional ASR and TTS models to enable speech input and output. These models showcase heightened intricacy, require substantial resources, and are susceptible to the inevitable issue of error accumulation. Recent works have started to explore end-to-end model architectures. SpeechGPT (Zhang et al., 2023) takes the discretized output of a speech model in self-supervised training and treats it as a specialized linguistic unit, training it alongside a large language model. However, due to the high sampling frequency of the discrete unit, it is difficult for this method to achieve multiple rounds of dialogue. LLaSM (Shu et al., 2023) has constructed an extensive speech instruction dataset intended for training the modal adapter to attain modality alignment. Their methodology is predominantly data-driven, with a lesser emphasis on the explicit design of modality alignment.

## 6 CONCLUSION

In this work, we introduce the BLSP approach, which bootstraps language-speech pre-training through behavior alignment of continuation writing. Our training procedure is straightforward, requiring only learning of a lightweight modality adapter through a novel utilization of speech recognition training data. As evidenced by quantitative evaluations in speech recognition, speech translation, spoken language understanding, and illustrated through multi-turn conversation demonstrations, BLSP effectively extends the remarkable language capabilities of LLMs to speech, enabling direct interaction with LLMs using speech input. Although there remains a substantial performance gap as well as numerous limitations (see Appendix D), BLSP represents a fresh and valuable perspective for achieving cross-modal alignment in LLMs, and there are numerous directions for expansion and improvement in future research.

## REPRODUCIBILITY STATEMENT

To increase reproducibility, we release our code at `https://anonymous.4open.science/r/blsp-4D8D`, which contains complete data processing scripts and training scripts. We will release our pre-trained BLSP model after the review process.

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

# A    INSTRUCTIONS USED FOR SLU TASKS

The instructions used for each SLU dataset are presented in Table 10.

| |
|---|
| **SNIPS:** *Please classify the intent of the text, choose from [DecreaseBrightness, IncreaseBrightness, SetLight-Brightness, SetLightColor, SwitchLightOff, SwitchLightOn].* |
| **FSC:** *Please classify the intent of the text, choose from [bring newspaper, deactivate lamp, change language English, deactivate music, increase heat, change language Korean, change language none, bring shoes, change language German, activate lights, bring socks, change language Chinese, decrease heat, decrease volume, increase volume, activate music, activate lamp, bring juice].* |
| **SLUE-VoxCeleb:** *Please classify the emotional tone of the text as either positive, negative, or neutral.* |

Table 10: Instructions used for SLU tasks.

## B  QUANTITATIVE ANALYSIS OF REPRESENTATIONS FROM BLSP

As depicted in Table 11, the representations of speech inputs learned from BLSP are distinct under various task instructions, unlike in Table 1 for ASR task. Table 12 illustrates the average cosine similarity between representations of paired speech and text inputs learned from BLSP, revealing a high level of similarity between the two, as opposed to the low similarity depicted in Table 2 for ASR task. We want to point out that there remains a notable gap between the representations for speech and text inputs that is worthy of future research.

|      | CW-S  | SA-S  | SR-S  | ST-S  |
|------|-------|-------|-------|-------|
| CW-S | 1.000 | 0.494 | 0.745 | 0.381 |
| SA-S | 0.494 | 1.000 | 0.501 | 0.278 |
| SR-S | 0.745 | 0.501 | 1.000 | 0.477 |
| ST-S | 0.381 | 0.278 | 0.477 | 1.000 |

Table 11: Average similarity between representations of the same speech inputs under different task instructions learned from BSLP.

| CW    | SA    | SR    | ST    |
|-------|-------|-------|-------|
| 0.785 | 0.866 | 0.808 | 0.900 |

Table 12: Average similarity between representations of paired speech/text inputs under the same task instructions learned from BLSP.

## C  SELECTED EXAMPLES OF CROSS-MODAL CONVERSATION

As demonstrated in Figure 7, BLSP provides expanded mechanisms to interact with LLMs. Users can freely switch between text and speech inputs, and directly employ speech instructions to carry out speech-to-text tasks.

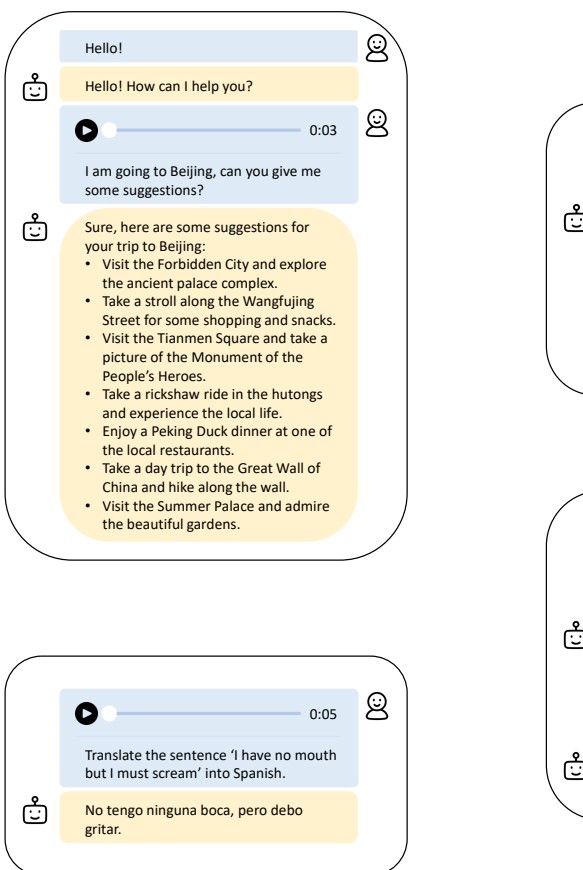

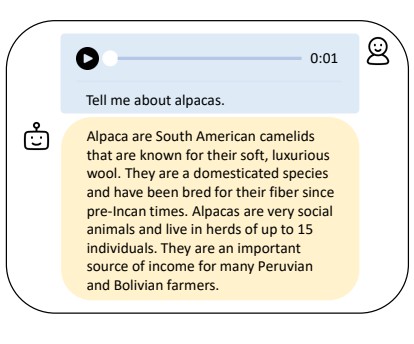

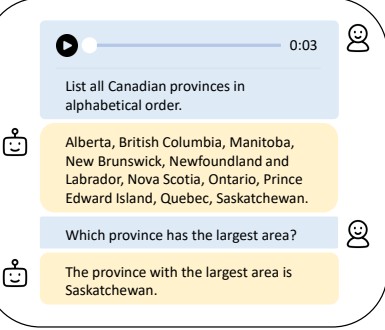

Figure 7: Selected examples of cross-modal conversation using BLSP model.

# D   LIMITATIONS

Although our BLSP approach can extend the remarkable language capabilities of LLMs to speech, as evidenced by quantitative evaluations and illustrative demonstrations, there are several limitations in our current study.

**Alignment Quality.**   As indicated by the quantitative evaluations, there exists a substantial performance gap when using speech input as opposed to the cascaded approach. Our approach to behavior alignment of continuation writing, in its current form, tends to align speech and text at a semantic level that restricts its capacity to capture detailed phonetic information. Exploring more fine-grained loss designs or approaches for constructing more fine-grained training data, including in combination with speech recognition, speech translation, or general speech instruction data, is worthy of further investigation.

**Paralinguistic Information.**   In this study, we mainly focus on aligning speech and text in the semantic space, without addressing the paralinguistic aspects of spoken language that cannot be simply described by words, such as emotions, tones, and intentions. It is possible to capture and incorporate paralinguistic information with LLMs by leveraging data from more diverse speech-related tasks, such as speaker identification, keyword spotting, and speech emotion recognition.

**Safety and Ethics.**   The use of continuous speech representations in our BLSP model could make it more susceptible to adversarial attacks and can potentially compromise the LLM's established adherence to the HHN criteria (Harmless, Helpful, Honest). This is an area that is worthy of future research, both in identifying weaknesses and searching for solutions.

**Broader Applicability.**   While our study has focused on the behavior alignment of continuation writing for speech-text alignment, the fundamental principles underlying this approach could have broader applicability. This involves expanding existing paired data in creative ways with the assistance of LLMs, ultimately benefiting LLMs. We leave it to future studies to extend this approach to diverse scenarios, including vision-language and multilingual alignments.

