# OpenReview forum: "BLSP: Bootstrapping Language-Speech Pre-training via Behavior Alignment of Continuation Writing"
_ICLR.cc/2024/Conference — Submitted to ICLR 2024_

### Official Review · Reviewer_Dd3G · 2023-10-29

**Soundness:** 3 good
**Presentation:** 3 good
**Contribution:** 4 excellent
**Rating:** 6
**Confidence:** 4

**Summary:**

The objective of the paper is to allow LLMs to be used with speech modality as input. Toward this, the authors propose to align the speech and text modalities through a method they call “continuation writing.” The key idea is to first use text input to generate text continuation, and then train an adaptor to predict the same continuation when the text is replaced with its corresponding speech input. The authors show that this simple strategy shows strong results on several tasks such as ASR, translation, and SLU. Analysis also reveals that this strategy helps to align the latent space of speech/text prompts for the same instruction, and push apart different instructions.

**Strengths:**

1. Arguably the biggest strength of the paper is the simplicity of the proposed method. There are no complex training strategies, nor the requirement of large amounts of hard-to-obtain data. The authors keep the largest components of the model (the speech encoder and the LLM) frozen and only train a small modality adapter, and this is done using publicly available ASR training data. As such, the BLSP method should be easy to replicate and use for downstream applications.

2. The authors have performed detailed empirical evaluation on several tasks: speech recognition (ASR), speech translation (ST), and spoken language understanding (SLU). While their model does not beat single-modality baselines (e.g., on ASR, the WERs are much worse than whisper-small, and on ST, the BLEU is worse than a cascade of ASR+LLM), it is still promising and reminds one of the early days of end-to-end speech translation. I believe with better modality adapters and training strategies, this method may get close to or surpass cascaded approaches.

3. More importantly, on the SLU task, BLSP outperforms a cascaded system since semantic similarity is sufficient for this task. This is an important take-away if this method is used to provide a common interface into a general-purpose QA system, which may only need to be semantically correct to provide accurate responses.

**Weaknesses:**

### Loss of speaker and paralinguistic information

Speech contains much more instruction than just its transcription, such as speaker, emotion, etc. Using paired ASR training data for modality alignment forgoes this extra information, and simply marginalizes over them. As such, the resulting model may only be good at tasks which worked with text inputs (i.e., recognition and semantic tasks). If we are extending an LLM with another modality, a natural requirement may be to extend the capabilities of the model — for example, for vision-language models, image understanding is usually achievable, which is not possible with text-only LLMs. But this kind of “extension” is not shown in this setting. It would have been great to evaluate the model on a non-semantic task (such as emotion recognition) which is not possible using text-only LLMs. This may be an adversarial task for this kind of training; in Section 3.1, the authors conjecture that “the LLM should behave the same no matter whether it is given the speech segment or its transcripts as input,” but this may not always hold.

### Regarding representations of modalities

In Section 2 (Figure 1), the authors find that training the modality adapter only on the ASR task is not beneficial since the input speech representations remain the same regardless of the task instruction used, meaning that the corresponding tokens are not being attended to. This is not surprising: the model only learns transcription since that is all it sees in training. Perhaps it would learn to attend to the instruction if the authors trained using a combination of different instructions, such as ASR, translation, and SLU, and also paraphrased the instructions themselves. The resulting model would still not be as good as BLSP, but would not see the representation collapse we observe in Figure 2.

Another common concern about joint speech-text training usually has to do with sequence lengths. Speech models usually need aggressive downsampling because of their large sequence lengths and redundancy of semantic information. For ASR, usually the speech-text length difference is solved using either cross-attention (e.g. LAS) or alignment-free training (e.g., CTC). For the BLSP model, the modality adapter takes the speech encoder representations, and transforms them for input to LLAMA, which is trained on autoregressive text decoding. Therefore, we expect the adapter to transform the speech representations to the same space as LLAMA input representations. In the paper, the authors used simple speech inputs not containing long pauses, so fixed-length subsampling worked well for them. However, in real scenarios, users often speak with pauses, or correct themselves, and so I wonder perhaps it would be a better idea to use variable-length subsampling techniques in the modality adapter [1].

[1] Y. Meng et al., "On Compressing Sequences for Self-Supervised Speech Models," 2022 IEEE Spoken Language Technology Workshop (SLT), Doha, Qatar, 2023, pp. 1128-1135, doi: 10.1109/SLT54892.2023.10022991.

### Empirical results

While the BLSP performance on ST and SLU is reasonable and promising, the degradation for ASR is quite large (Table 3). In particular, LLAMA may already have seen the audiobooks from LibriSpeech, or transcripts of the TED talks from TED-LIUM 3, so I expected these WERs to be much lower. The authors suggest (in Table 4) that this is mainly because of the nature of LLAMA to produce fluent text output (probably because of the prompt used). I have two questions about this:
1. Can the authors show WER results only on rare words to see if important words are transcribed faithfully (without caring about stop words)?
2. Did the authors try different prompts to see what is the lowest WER that can be achieved? For example, from an application perspective, it may be very useful to have one model that can generate both verbatim and non-verbatim transcripts simply by using different prompts.

**Questions:**

1. Is the term “continuation writing” used before? Why did the authors not simply call it “next token prediction”?
2. In Section 4.4, the authors show multilingual capabilities of the trained model even though the modality adapter is only trained with English ASR data. This suggests that the transformation from speech representations to corresponding text is the same (or similar) across all languages. Can the authors measure this isomorphism among spaces more carefully, perhaps using some of the techniques in [2]?

[2] Marchisio, Kelly et al. “IsoVec: Controlling the Relative Isomorphism of Word Embedding Spaces.” ArXiv abs/2210.05098 (2022): n. pag.

---

> ### Author Response · Authors · 2023-11-16
>
> We appreciate your accurate assessment of the strengths of our paper and your insightful comments about its weaknesses. Before responding to your specific questions, we would like to highlight your observation regarding the use of our method in a general-purpose QA system. In Section 4.3, we demonstrated through cross-modal conversation demos that our model can process general-purpose speech instructions and generate accurate responses. A video demonstration showcasing this capability is available at [https://anonymous4blsp.github.io/iclr24-3521.github.io/](https://anonymous4blsp.github.io/iclr24-3521.github.io/), as referenced in our paper.
>
> ### Q1. Loss of Speaker and Paralinguistic Information
>
> As discussed in Appendix D and referenced in the conclusion section, one of the limitations of our current method is the loss of speaker and paralinguistic information. Our focus is on aligning the semantic content of speech and text within the context of LLMs, a challenge that remains a focus of active research in the community [1,2,3]. Consequently, in the scope of our study, the statement "the LLM should behave the same, whether given the speech segment or its transcripts as input" holds true when considering semantic content alone.
>
> However, our approach provides a pretrained end-to-end speech-text LLM with potential adaptability to capture speaker and paralinguistic information through fine-tuning on appropriate training data. Our analysis in Table 8 of Section 4.2 demonstrates that our model can be more effectively fine-tuned for downstream tasks (ST in this case) compared to models without pretraining or those pretrained with ASR. We plan to explore ways to capture speaker and paralinguistic information in future work.
>
> \[1\] Zhang, Dong, et al. "Speechgpt: Empowering large language models with intrinsic cross-modal conversational abilities." _arXiv preprint arXiv:2305.11000_ (2023).
>
> \[2\] Shu, Yu, et al. "Llasm: Large language and speech model." _arXiv preprint arXiv:2308.15930_ (2023).
>
> \[3\] Wang, Mingqiu, et al. "SLM: Bridge the thin gap between speech and text foundation models." _arXiv preprint arXiv:2310.00230_ (2023).
>
> ### Q2. Representations of Modalities
>
> **A. Representation Collapse**
>
> The occurrence of mode collapse in speech features resulting from ASR training is not immediately evident. In computer vision, BLIP2 [4] has shown that training solely on image caption data can enable general instruction-following capabilities for zero-shot image-to-text generation tasks. Similarly, in the field of speech input, pretraining the model using the ASR task has been a common practice in prior work [1,2]. However, the benefits and limitations of this approach were not thoroughly studied.
>
> We have explored different methods to utilize the ASR task for enabling instruction-following capabilities. As described in footnote 2, we used GPT4 to generate 100 diverse prompts for the ASR task, yet representation collapse still occurred. We also experimented with training on both English ASR and English-to-Chinese speech translation (ST) tasks using respective instructions. However, the resulting model still failed to demonstrate instruction-following capabilities for unseen tasks. Generally, training on specific downstream tasks tends to limit the model's abilities to those particular tasks, making it challenging to generalize to unseen instructions. This phenomenon has also been highlighted in a recently released work [5]. Additionally, we want to emphasize that labeled speech-text training data is quite limited outside of ASR and ST.
>
> \[4\] Li, Junnan, et al. "Blip-2: Bootstrapping language-image pre-training with frozen image encoders and large language models." _arXiv preprint arXiv:2301.12597_ (2023).
>
> \[5\] Pan, Jing, et al. "COSMIC: Data Efficient Instruction-tuning For Speech In-Context Learning." _arXiv preprint arXiv:2311.02248_ (2023).
>
> **B. Choice of Fixed-Length Subsampling**
>
> Thank you for highlighting this important aspect and suggesting the exploration of dynamic downsampling strategies. We recognize that different downsampling methods can impact the model's performance, and there are indeed various design options to consider. In this study, our primary focus was on investigating the instruction-following capability for speech input. For this purpose, we chose a simple yet efficient approach, employing a CNN followed by a bottleneck layer as the adapter structure.
>
> We are currently exploring more sophisticated methods for the adapter, which could potentially enhance the model's performance further. However, this exploration is part of our ongoing research and will be detailed in a future publication.

---

> ### Author Response · Authors · 2023-11-16
>
> ### Q3. Empirical results
>
> **A. WER on Rare Words**
>
> We acknowledge that a well-aligned speech-text LLM model should ideally perform ASR tasks with a lower WER, and we recognize that a 20% WER might seem high. In our experiments, the WER remains approximately 20% even after removing stop words. This outcome can be attributed to our training approach, which emphasizes semantic alignment over lexical precision. For instance, phrases like "I love cats" and "I like cats" would generate the same continuation in our model, as the adapter isn’t explicitly trained to capture fine-grained lexical details.
>
> In our continuing research, we have found that incorporating a modest amount (approximately 10%) of ASR data alongside our continuation writing data can notably enhance the WER, reducing it to about 4-5%, while simultaneously preserving the model's broad instruction-following capabilities. However, it is important to highlight that this paper primarily concentrates on developing methods to facilitate general instruction-following abilities. Consequently, we have intentionally chosen not to optimize performance for any specific task.
>
> **B. Impact of Different Prompts on WER**
>
> You are indeed correct in noting that different instructions can influence WER. As highlighted in footnote 4, employing the prompt _"Please repeat the following words."_ results in lower WER scores—17.0 on LibriSpeech-dev and 17.4 on LibriSpeech-test. However, our current method aligns speech and text at a semantic level. Consequently, the adapter cannot capture detailed lexical information, making it difficult to address this issue merely through changes in prompts.
>
> ### Q4. 'Continuation Writing' vs. 'Next Token Prediction'
>
> The term 'next token prediction' applies broadly to a variety of LLM training scenarios, including those that involve the ASR task. In such cases, the model optimizes the prediction of the reference transcript through 'next token prediction,' utilizing both textual instructions and speech features. However, we have chosen to use the term 'continuation writing' in our work to specifically distinguish our training task from traditional 'ASR' or other similar tasks.
>
> ### Q5. More detailed analysis on the multilingual space.
>
> Thank you for the suggestion to quantitatively measure isomorphism.
>
> We selected 1,000 quadruples (English text, English speech, French text, French speech) from the CVSS test set. We extracted text features at LLM input via word embedding and obtained speech features from the output of either the Whisper encoder or our modality adapter. Sentence embeddings were then computed through average pooling, allowing us to calculate the relational similarity.
>
> |       | **LLM input** | **Whisper Output** | **Adapter Output** |
> | ----- | ------------- | ------------------ | ------------------ |
> | en-fr | 0.60          | 0.44               | 0.39               |
>
> As indicated in the table, the multilingual space of LLM inputs shows strong isomorphism, as does the space of Whisper outputs. Even when using only English ASR data for learning the transformation, the adapter’s output space retains the isomorphism characteristic of Whisper.

---

> > ### Comment · Reviewer_Dd3G · 2023-11-20
> >
> > Thanks for your reply to my comments. It is nice to see that task-specific training improves performance (e.g. WER) without compromising on the general instruction-following ability.

---

### Official Review · Reviewer_tfEL · 2023-10-29

**Soundness:** 1 poor
**Presentation:** 2 fair
**Contribution:** 1 poor
**Rating:** 5
**Confidence:** 2

**Summary:**

This paper proposes training an adapter to connect an audio encoder to a text LLM so that speech tasks such as ASR, sentiment analysis, translation, and continuation writing can be performed using the same setup. Initial experiments with frozen Whisper encoder and Llama2 LLM does not show an improvement on the tasks. In Section 4.2. where there is fine-tuning of the models, there is some improvement on the translation task (Table 8).

**Strengths:**

Originality:
The paper puts simple ideas together to achieve multiple speech tasks in a single model which can also extend to other languages.

Quality:
1. Analysis in Section 2 on overtuning for ASR and not generalizing well to other speech tasks of models without adapter is useful.

Clarity:
Limited at times. The goal of the paper is not always clear.

Significance:
Combining speech and LLMs is getting popular as it was discussed in Section 5.

**Weaknesses:**

Experiments do not show positive results especially when the audio encoder and the LLM is freezed. The goal of the paper is not clear to the reviewer.
- If the main contribution is the way the paper prepares the data, then it is not the only factor for success given the experimental results.
- If the goal is to show the effect of fine-tuning, it provides some marginal gain.
- If the goal is to show the usefulness of behavioral alignment, it seems as if giving "noisy" input in the sense that an audio, a text prompt and some other irrelevant (?) text and let the model ignore the continuation text and perform the recognition task. I might be misunderstanding this, but it sounds as if the experiment is doing some robustness improvement on the model with this type of "noisy" data.

**Questions:**

1. Please clarify the goal of the experiments more clearly.

2. Is having a BLUE score over 5 good enough in terms of model quality given that the baseline models achieve over 15? Similarly, a WER over 20% does not show the benefit of the model.

3. Section 4.2 shows some minor improvements on the speech translation task. Are there similar experiments on other tasks discussed earlier in the paper?

---

> ### Author Response · Authors · 2023-11-16
>
> Thank you for your feedback. It appears there may be some misunderstanding regarding the motivation, contributions, and experimental design of our work. We encourage you to refer to the "General Response" provided above and the point-by-point response below for clarifications. We sincerely hope that these clarifications will be helpful for your reassessment of our research.
>
> ### Q1. The Goal of The Paper
>
> While we believe that an end-to-end approach can eventually outperform a pipeline approach with expanded capabilities (e.g., using paralinguistic information), that is not the immediate goal of this paper. As explained in the "General Response," our objective is to extend the instruction-following capability of LLMs to speech input to enable zero-shot capabilities for unseen tasks in an end-to-end model.
>
> In Section 4.1, we demonstrate this by using textual instructions to prompt the model for unseen speech-to-text generation tasks. For instance, our model, trained solely on continuation tasks, can be prompted during inference for ASR tasks as follows:
>
>     ###[Human]: Please transcribe the following speech into text. [speech features]
>     ###[Assistant]: [response]
>
> Here, the ASR instruction can be substituted with instructions for other tasks, such as ST and SLU tasks evaluated in the paper, or tasks like summarization and question-answering. We emphasize that our goal in this paper is not to outperform the pipelined approach, but to demonstrate the model's ability to follow instructions across modalities in an end-to-end manner. The comparison with the pipeline approach is to both highlight the promising results of the end-to-end approach and the performance gap with the pipeline approach, encouraging future research to further reduce this gap and eventually surpass it. This aspect is discussed in the limitations of the current approach in the Appendix and referenced in the conclusion section.
>
> In Section 4.2, we present analyses to highlight different aspects of our model. However, these should not be viewed as the main experiment. Table 8 illustrates that our method allows for effective fine-tuning on downstream tasks. Figure 3 demonstrates successful speech-text modality alignment, while Figure 4 investigates the impact of data scale on alignment effectiveness.
>
> ### Q2. Performance Gap with Baseline Models
>
> As explained in the response to Q1, the focus of our approach is on cross-modal instruction-following capabilities, rather than on the improvement of performance in specific tasks. Similar to the early days of end-to-end speech translation, our approach is not immediately expected to surpass the performance of ASR+LLM or Text+LLM models. Our primary contribution lies in proposing an end-to-end training methodology for speech-text LLMs, thereby demonstrating the model's valuable ability to follow instructions across modalities, despite the current performance gap. We hope our work will be beneficial for the research community in progressing towards the ultimate goal of developing more effective end-to-end approaches.
>
> ### Q3. Minor Improvement in Table 8
>
> As noted in response to Q1, our main focus in this paper is on the instruction-following capabilities for speech input, as presented in quantitative zero-shot evaluations in Section 4. The analyses in Section 4.2 are designed to highlight different aspects of our model. Specifically, Table 8 in Section 4.2 is intended to demonstrate that our pretrained model is effective not only in its instruction-following capabilities but also for fine-tuning on downstream tasks. The improvement over models without pretraining is substantial in both BLEU and COMET metrics, and is significantly better than ASR pretraining, with an average improvement of 0.5 BLEU and 1 COMET score across eight directions. However, we wish to emphasize that this aspect of the model is not the main focus of our paper.

---

> > ### Comment · Reviewer_tfEL · 2023-11-22
> > **Thanks for clarification**
> >
> > I would like to thank the authors for providing clarification.
> > - Regarding the goal, I see that the goal is to demonstrate the text interpretation capabilities of LLM with audio - text alignment.
> >
> > - Q2, even though an initial attempt to achieve modality alignment in an end-to-end fashion (as compared to the ASR + LLM cascade) is challenging by itself and the results in Tables 3, 5, 6, 7 show decent performance, However, there is still a significant performance gap between the ASR-based and encoder based solutions.
> >
> > - As for Q3, I understand that Table 8 and 9 are additional experiments, and Table 8 shows some benefit of finetuning. However, the results in Table 9 are significantly worse than the baselines and hence, it is hard to say that the model has multilingual multilingual capacities.
> >
> > I admit that the study is a valuable effort and there are some signs that it is going to work better with further improvements. As a result, I am increasing my score to marginally below the threshold.

---

> > > ### Author Response · Authors · 2023-11-23
> > >
> > > Thank you again for taking the time to read our response and share your feedback. Your concerns regarding the performance gap are valid; however, we would like to take this opportunity to share our perspective on why we believe the performance gap should not negate the contributions of this paper.
> > >
> > > ### Q1. Performance Gap with Cascaded Models
> > >
> > > As we acknowledged in the paper, there currently exists a performance gap between our proposed end-to-end approach and the pipelined approach. We want to note that the primary contribution of our work is to demonstrate the effectiveness of the training method in transferring instruction-following capabilities from text-based language models (LLMs) to the speech modality. We deliberately compare with the pipelined approach to highlight both the potential of our approach as well as its current limitations.
> > >
> > > Developing end-to-end LLMs with the ability to follow cross-modal instructions for speech input is not trivial and has become an active research area [1,2,3,4]. While these studies focused on constructing new speech instruction datasets, our work provides a new approach through behavior alignment and takes advantage of existing ASR training data. In contrast to most recent works that demonstrate their effectiveness through demos, we provided both demonstrations and quantitative evaluations to understand both the strengths and limitations. It is our hope that this transparent approach will provide the research community with useful insights to continue developing new approaches to reduce the performance gap and ultimately surpass pipelined approaches.
> > >
> > > We also want to mention the similarity to the development of end-to-end approaches to other problems. The concept of end-to-end speech translation was initially proposed in 2016 with a significant performance gap compared to cascaded systems. It was not until 2020 that end-to-end speech translation achieved comparable performance to cascaded systems and eventually became the mainstream approach. The research efforts and publications during these years have collectively contributed to the advancement of the research field.
> > >
> > > Finally, we want to acknowledge again that more research is needed to further improve the performance of the end-to-end approach, and some of our ongoing research has achieved  improvement in this direction, as we also noted in part in responses to other reviewers. However, we believe the approach developed in this paper has its own merit and would be beneficial to the research community.
> > >
> > > [1] Zhang, Dong, et al. "Speechgpt: Empowering large language models with intrinsic cross-modal conversational abilities." arXiv preprint arXiv:2305.11000 (2023).
> > >
> > > [2] Shu, Yu, et al. "Llasm: Large language and speech model." arXiv preprint arXiv:2308.15930 (2023).
> > >
> > > [3] Wang, Mingqiu, et al. "SLM: Bridge the thin gap between speech and text foundation models." arXiv preprint arXiv:2310.00230 (2023).
> > >
> > > [4] Pan, Jing, et al. "COSMIC: Data Efficient Instruction-tuning For Speech In-Context Learning." arXiv preprint arXiv:2311.02248 (2023).
> > >
> > > ### Q2. The Multilingual Capability of BLSP
> > >
> > > In Section 4.5, our objective is to demonstrate that our model can exhibit a certain level of understanding of non-English speech inputs, despite being trained solely on English data. We consider this a nontrivial result that highlights the potential of this approach, and we compare it with the pipelined approach to demonstrate its strengths and limitations. Again, we believe that this quantitative analysis does not undermine the contributions and conclusions drawn in our study, and we hope it is helpful for a better understanding of the approach and will be beneficial for the research community.

---

### Official Review · Reviewer_EDCC · 2023-10-31

**Soundness:** 4 excellent
**Presentation:** 4 excellent
**Contribution:** 4 excellent
**Rating:** 8
**Confidence:** 3

**Summary:**

This work introduces BLSP, which attempts to align speech and text modality in LLM. The model first starts collect supervised samples from LLM by generating text continuation based on speech text via instruction. Those supervised samples are then used to train a modality adapter on Whisper encoder, which helps align speech and text modality in LLM generation.

The experiment demonstrates that it can achieve unseen tasks to some extent even those the adapter was only trained with the continuation task. Although some tasks are not as good as the cascade systems, it shows some promising results in some tasks (i.e. speech understanding). Further analysis demonstrates that the text and speech embedding are better aligned using this approach and it also demonstrates some capabilities across languages.

**Strengths:**

This work proposes to align speech and text modality in LLM and successfully show the proposed protocol allows the model to achieve unseen tasks even it is only trained with the continuation task. I think it has lots of potential for this direction.

The experiment analysis over a few speech tasks are convincing and demonstrate its usefulness, especially in the speech understanding task

**Weaknesses:**

The speech encoder is from Whisper-small which has only 120M parameter (244M/2 as it only uses encoder) , this is considerably much smaller than the LLM (7B). Using it as models/baselines might not be strong enough, although it might because it is bound by the large GPU memory caused by LLM.

The results of ASR/translation task still has a large gap with the text-based model.

**Questions:**

In Figure 1, only 1 type (grey triangle) of speech embedding is plotted, where is the other speech embeddings? are they overlapped with other symbols?

The convolution modality reduces the length of the speech features by a factor of 8, how did authors choose this reduction factor? does author also try changing this factor? for instance, larger reducing factor might reduce lexical info, but make semantic info denser.

---

> ### Author Response · Authors · 2023-11-16
>
> Thank you for recognizing the value of our work and for your insightful comments. Below are our responses to your questions.
>
> ### Q1. Choice of Speech Encoder & Performance Gap with Text-Based Model
>
> We acknowledge that using Whisper-small as the speech encoder might not represent the strongest baseline. However, our primary focus is not on the model's performance on downstream tasks, but rather on demonstrating our training method's ability to transfer instruction-following capabilities from text-based LLMs to the speech modality. As you pointed out, we opted for Whisper-small due to computational considerations during the development of our approach. However, we did conduct some experiments (not included in the paper) with Whisper-large models and observed improved zero-shot performance on unseen tasks. These experiments, however, did not alter the paper's primary conclusions.
>
> As highlighted in Appendix D and referenced in the conclusion section, the performance gap compared to the pipelined approach remains a limitation of our current approach. We are actively exploring various strategies to bridge this gap and to expand the model's capabilities. It is our hope that the methodology proposed in this paper will contribute to the research community's efforts toward achieving these goals.
>
> ### Q2. Overlapping Speech Embeddings
>
> You are correct in your observation. Models trained with ASR data tend to lose their ability to follow diverse instructions. When different instructions are used to prompt speech input, nearly identical features are produced, leading to the overlapping of variously colored triangles in Figure 1. We will make this point clear in the revision.
>
> ### Q3. Choice of 8x Reduction Factor
>
> We agree that a larger reduction factor could diminish lexical information, as demonstrated in [1]. However, a smaller reduction factor would result in longer speech features, excessively consuming context in the LLM. This could decrease training and inference efficiency and limit the number of rounds in multi-turn interactions. In our empirical studies, we observed that 1 second of speech corresponds to approximately 3-4 text tokens. Given that the output frequency of the Whisper encoder is 50Hz, we chose an 8x downsampling factor to balance training efficiency with the preservation of lexical information. We believe that choosing a larger or smaller reduction factor would not alter the main conclusions of the paper.
>
> \[1\] Fathullah, Yassir, et al. "Prompting large language models with speech recognition abilities." _arXiv preprint arXiv:2307.11795_ (2023).

---

> > ### Comment · Reviewer_EDCC · 2023-11-21
> >
> > Thank you for your comments, it is interesting to see a stronger whisper model can indeed improve the performance.

---

### Official Review · Reviewer_ZRsu · 2023-11-01

**Soundness:** 2 fair
**Presentation:** 3 good
**Contribution:** 3 good
**Rating:** 6
**Confidence:** 4

**Summary:**

The paper proposes a speech-text modality alignment method based on learning a lightweight modality adapter by continuation writing using continuations generated from LLM and speech transcript as supervised signals. Compared to ASR task-based pre-training, the proposed method gives good alignment and better speech translation performance.

**Strengths:**

The paper proposes a pre-training method for a lightweight modality adapter by continuation writing, which works better than ASR task-based pre-training.

**Weaknesses:**

The advantage of the proposed method is unclear from the experiments.

**Questions:**

Is it correct that if the modality adapter does nothing but output the input obtained from the encoder as it is (or learns an identical transformation), high alignment is obtained since you use an ASR system as the encoder?

How did you choose the structure of the adapter?

In the experiment in Table 8, where you update the speech encoder, what is the performance of the cascade approach with the fine-tuning of the ASR module?

What is the performance if you use other speech encoders?

---

> ### Author Response · Authors · 2023-11-16
>
> Thank you for your feedback. It appears there may be some misunderstanding regarding the motivation, contributions, and experimental design of our work. We encourage you to refer to the "General Response" provided above and the point-by-point response below for clarifications. We sincerely hope that these clarifications will be helpful for your reassessment of our research.
>
> It's important to note that while we included studies on ASR task-based pretraining, our goal was to demonstrate the inadequacy of ASR task-based pretraining for our purpose, as models trained in this manner cannot follow general instructions at all. Therefore, in all experiments evaluating zero-shot instruction capabilities – which are central to our study – we do not compare our results with those of ASR task-based pretraining. The comparison with ASR task-based pretraining is made only in the fine-tuning experiment in Section 4.2. This is to illustrate that BLSP is a more effective pretraining method for fine-tuning on downstream tasks.
>
> ### Q1. Use of ASR System as the Speech Encoder?
>
> We believe there is a misunderstanding that we would like to clarify. The Whisper ASR system is comprised of an encoder for acoustic feature extraction and a decoder for text generation. In our work, we utilized only the pretrained Whisper encoder as the speech encoder for extracting acoustic features, while discarding the Whisper decoder. Our proposed model functions as an **end-to-end** system, where the speech encoder's role is to extract features from the input speech, rather than to transcribe the speech into text. Including a Whisper decoder, as used in a standard ASR system, would necessitate an argmax operation, which would impede the feasibility of training this model in an end-to-end manner. Additionally, this would introduce the limitations of a pipelined approach, as discussed in our "General Response."
>
> The modality adapter plays a significant role in aligning speech features with the textual space, and it does not merely perform an identical transformation. Learning a mapping that effectively transfers the instruction-following capability from text to speech input is a nontrival task. As demonstrated in Section 2, a modality adapter trained solely on the ASR task lacks general instruction-following capabilities.
>
> ### Q2. Choice of Adapter Structure
>
> The modality adapter in our study serves two primary functions. First, it reduces the length of output from the Whisper encoder, which operates at a frequency of 50Hz. Second, it maps the speech feature space to the textual space of the LLM. We acknowledge that various structural options exist for the adapter. For length reduction, methods like CNN downsampling, frame stacking, or q-transformer are potential choices. Similarly, for cross-modality alignment, both MLP and transformer-based methods are feasible.
>
> However, our study's focus was not on exhaustively exploring different adapter structures. Instead, our main contribution lies in proposing a novel training method to connect a frozen speech encoder with an LLM. Consequently, we opted for a simple yet effective approach, employing a CNN followed by a bottleneck layer for the adapter. This decision allowed us to concentrate on the core aspect of our research while maintaining effectiveness in achieving our objectives.
>
> ### Q3. Performance of Fine-Tuned ASR Module in a Cascaded System
>
> BLSP is designed as an end-to-end speech-text model. In the experiments outlined in Table 8, we fine-tuned the entire model using speech translation (ST) data via an end-to-end instruction-tuning approach:
>
>     ###[Human]: Please translate the following speech into German. [speech features]
>     ###[Assistant]: [translation]
>
> During this process, the Whisper encoder was fine-tuned along with other model parameters, but the corresponding Whisper decoder was not used or fine-tuned. Consequently, it's not feasible to measure the ASR performance of the fine-tuned Whisper encoder as would be done in a traditional Whisper ASR system. Moreover, since the ST data comprises (speech, translation) pairs rather than (speech, transcription) pairs, it does not lend itself to optimizing a separate Whisper ASR system. Therefore, we are unable to report the performance of a cascaded system using a fine-tuned ASR module.
>
> The purpose of Table 8 is to show that our proposed approach, in comparison to direct training on ASR data, offers better generalization and is more suitable for fine-tuning on downstream tasks. However, we emphasize that this is not the main contribution of our work. Our primary contribution, as detailed in Section 4.1, is the demonstration of the cross-modal instruction-following capability of our model and its ability to generalize to unseen tasks through text instructions.

---

> ### Author Response · Authors · 2023-11-16
>
> ### Q4. Using a Different Speech Encoder?
>
> We acknowledge that there are various options for the speech encoder, such as the Whisper model's encoder [1], USM [2], or other pre-trained speech models, which could impact the model's performance. However, our work  focuses on the methodology of aligning a frozen speech encoder with a frozen LLM to achieve instruction-following capability for speech input on unseen tasks. As such, an in-depth investigation into different speech encoders is not the primary focus of this work, and we believe that changing the speech encoder would not fundamentally alter the conclusions of our paper.
>
> We chose the encoder of the Whisper model due to its widespread recognition and to facilitate comparison with a strong ASR model in the pipelined approach. This choice mirrors our decision to use Llama2 as the LLM, despite the availability of many open-source LLM options. Our aim was to employ well-known and robust components to showcase the effectiveness of our methodology.
>
> \[1\] Shu, Yu, et al. "Llasm: Large language and speech model." arXiv preprint arXiv:2308.15930 (2023).
>
> \[2\] Wang, Mingqiu, et al. "SLM: Bridge the thin gap between speech and text foundation models." arXiv preprint arXiv:2310.00230 (2023).

---

### Author Response · Authors · 2023-11-16

## General Response

We appreciate your insightful feedback on our paper. The varied opinions among the reviews suggest some confusion about our work's motivation, contributions, and experimental design, which we would like to clarify before addressing specific comments.

The primary goal of our research is to extend the instruction-following capabilities of text-based Large Language Models (LLMs) to speech inputs within an end-to-end model. The standard pipelined approach, combining ASR systems with LLMs, faces challenges such as error accumulation, increased inference time, and limited access to acoustic features that could enhance speech understanding and enable the exploration of paralinguistic information.

In response, developing end-to-end LLMs for speech inputs has become an active research area. Recent approaches include using synthesized speech instruction data from text instructions [1], which lack diversity in speech signals, and collecting new speech instruction data [2], a process that is difficult and resource-intensive. These methods, while providing demonstrative examples of instruction-following capabilities, often lack quantitative evaluations, partially due to the aforementioned limitations.


Our novel approach leverages ASR training data to create unique continuation writing instructions. We have shown that a lightweight modality adapter, used in conjunction with a frozen acoustic encoder and LLM, can effectively align speech representations to the LLM. This alignment allows the model to follow cross-modal instructions, a capability not extensively explored in previous studies. We clarify that the Whisper encoder in our approach serves as an acoustic encoder for extracting speech representations, not as a complete ASR system. The modality adapter's role is critical in aligning speech representations to the LLM's word embedding space and addressing the length disparity

It is important to emphasize that our immediate goal is not to surpass the performance of pipelined approaches. Our aim is to propose a new methodology for developing an end-to-end model that links pretrained speech and text models. Our zero-shot experiments demonstrate the model's cross-modal instruction-following capabilities, and our speech translation (ST) finetuning experiment highlights the potential for further fine-tuning on speech instruction or other speech-text training data. We anticipate that continued research in this direction will lead to improved performance, eventually outperforming the pipeline approach and expanding capabilities, including the integration of paralinguistic information.

We acknowledge the limitations of our current approach, detailed in the Appendix and referenced in the conclusion section, and are actively exploring ways to enhance the model's performance and capabilities. We hope this response has clarified our work's intentions and contributions.

\[1\] Zhang, Dong, et al. "Speechgpt: Empowering large language models with intrinsic cross-modal conversational abilities." _arXiv preprint arXiv:2305.11000_ (2023).

\[2\] Shu, Yu, et al. "Llasm: Large language and speech model." _arXiv preprint arXiv:2308.15930_ (2023).

---

### Comment · Area_Chair_ZUGK · 2023-11-21
**Reminder to reviewers to participate in the author/reviewer discussion**

Dear reviewers, this is a reminder that the author/reviewer discussion period ends November 22.

This discussion is indeed supposed to be a dialog, so please respond to
the comments from the authors.

@Reviewer Dd3G and @Reviewer EDCC - thank you for responding to the authors.

AC

---

### Meta-Review · Area_Chair_ZUGK · 2023-12-06

**Metareview:**

## Scientific Claims and Findings
This paper addresses the question of how to enable a large, text-based language model to also process speech inputs in a manner that enables end-to-end training. The approach explored in this paper is to begin with a strong pre-trained speech encoder (the **encoder** portion of the Whisper-small model) and a strong language model (Llama-2-7B, fine-tuned on Alpaca-52K to enable instruction-following), and then to train a lightweight modality adapter that converts the output of the speech encoder into acoustic tokens that are input to the LM. The paper begins with the observation that if this adapter is trained using an automatic speech recognition criterion, in which the LM is trained to produce the transcripts corresponding to sequences of acoustic tokens, the adapter fails to align the speech and text modalities: the representations of cross-modal speech+text prompts remain distinct from the representations of text-only prompts, and the cross-modal representations are not much influenced by textual instructions. To address this difficulty, the paper proposes using continuation writing as the training task for the adapter instead of using automatic speech recognition. The LM is first prompted to write a continuation of each textual transcript from the ASR training data. Then, the encoder, adapter, and LM are trained to generate these continuations given a "continuation prompt" and the acoustic features. In this process, the encoder and LM are kept frozen. Experimental results show that the resulting model is able, to some extent, to perform speech recognition, speech translation, and spoken language understanding in a zero-shot fashion, that the continuation writing task greatly reduces the speech/text modality gap, and that some alignment of the speech and text modalities across languages is achieved, even though the continuation writing task is performed only in English.

## Strengths
- The proposed continuation writing task for training the modality adapter clearly outperforms the automatic speech recognition task in this set of experiments.
- The continuation writing task can be readily implemented and used by other researchers.

## Weaknesses
- The performance of the model on downstream tasks is relatively poor, which makes the paper less convincing.
- Based on the discussions, the authors appear to have some methods available to them to improve downstream task performance such as using a stronger acoustic encoder and mixing some ASR training in with the continuation writing training, but they did not add these to the paper.
- The initial experiment with the ASR task for training the modality adapter is not described in detail. Is exactly the same architecture used as in the continuation writing experiments? What losses are achived by the model for the ASR and continuation writing tasks? Do the two tasks lead to roughly the same amount of supervision (in other words, how many tokens are predicted in the ASR task and how many in the continuation writing task)? Even if there is not enough space in the main paper to provide this information, it is important enough to the understanding of the paper that it should at least be provided in the supplementary materials.

**Justification For Why Not Higher Score:**

- The continuation writing task definitely appears to have potential for bridging the modality gap between speech and text, but the paper would be a lot more convincing if there were stronger performance on the downstream tasks.

- The point raised by Reviewer Dd3G that the LLM may have already seen the text used in the ASR tasks is an important one that really ought to be addressed.

**Justification For Why Not Lower Score:**

N/A

---

### Decision · Program_Chairs · 2024-01-16

Reject